# Assessing Africa's child survival gains and prospects for attaining SDG target on child mortality

Sunday A. Adedini[1,2]*, Seun Stephen Anjorin[3], Jacob Wale Mobolaji[4], Elvis Anyaehiechukwu Okolie[5,6], Sanni Yaya[7,8]

1 Department of Demography and Social Statistics, Federal University Oye-Ekiti, Oye-Ekiti, Nigeria, 2 Programme in Demography and Population Studies, Schools of Public Health and Social Sciences, University of the Witwatersrand, Johannesburg, South Africa, 3 Nuffield Department of Population Health, Big Data Institute, University of Oxford, Oxford, United Kingdom, 4 Department of Demography and Social Statistics, Obafemi Awolowo University, Ile-Ife, Nigeria, 5 School of Health and Life Sciences, Teesside University, Middlesbrough, United Kingdom, 6 Department of Public Health, David Umahi Federal University of Health Sciences, Uburu, Ebonyi State, Nigeria, 7 Faculty of Social Sciences, School of International Development and Global Studies, University of Ottawa, Ottawa, Ontario, Canada, 8 The George Institute for Global Health, Imperial College London, London, United Kingdom

* Sunday.Adedini@fuoye.edu.ng

**Data Availability Statement:** Datasets for this study were obtained from the World Development Indicator, WHO's Global Health Observatory Data, Demographic and Health Survey (DHS), and

## Abstract

This study assessed Africa's child survival gains and prospects for attaining Sustainable Development Goals (SDG) target 3.2. We analysed multiple country-level secondary data-sets of 54 African countries and presented spatial analysis. Results showed that only 8 out of the 54 African countries have achieved substantial reductions in under-5 mortality with an under-five mortality rate (U5MR) of 25 deaths per 1,000 live births or less. Many countries are far from achieving this target. Results of the predictions using supervised machine learning on the Bayesian network reveal that the probability of achieving the SDG target 3.2 (i.e., having U5MR of 25 deaths per 1000 live births or less) increases (from 21.6% to 100%) when the contraceptive prevalence increases from 49.8% to 78.5%; and the use of skilled birth attendants increases from 44.8% to 86.3%; and percentage of secondary school completion of female increases from 42.5 to 74.0%. Our results from Local indicator of spatial autocorrelation (LISA) cluster maps show that 7 countries (mainly in West/Central Africa) formed the high-high clusters (hotspots for U5M) and may not achieve the SDG target 3.2 unless urgent and appropriate investments are deployed. As 2030 approaches, there is a need to address the problem of limited access to quality health care, female illiteracy, limited access to safe water, and poor access to quality family planning services, particularly across many sub-Saharan African countries.

## Introduction

There have been considerable gains in reducing the mortality risks of children below the age of five globally. The last three decades have witnessed a 58% decrease in global child mortality,

Human Development Report (HDR). These are publicly available data that can be accessed upon request at https://dhsprogram.com/Data/, https://www.bing.com/search?q=World+Development+Indicator&qs=ds&form=QBRE, and https://www.who.int/data/gho Others would be able to access these data in the same manner as the authors. We also confirm that the authors did not have any special access privileges that others would not have.

**Funding:** The authors received no specific funding for this work.

**Competing interests:** The authors have declared that no competing interests exist.

from 93 deaths per 1000 live births in 1990 to 39 deaths per 1000 live births in 2017 [1]. Child mortality rate is among the best indicators of child health and, more generally, of social and economic development in low- and middle-income countries (LMICs). Child mortality is measured using the under-five mortality rate (U5MR), which is defined as the probability of a child dying between birth and the fifth year of life [2]. The Millennium Development Goals (MDGs) adopted this measure as the primary target for MDG 4 which sought to reduce U5MR by two-thirds between 1990 and 2015 [3,4]. This target was monitored via three indicators: under-five mortality, infant mortality, and the proportion of one-year-old children immunised against measles.

The MDGs mobilised global efforts towards the promotion of child survival, hence the successful decline in child mortality worldwide by half between 1990 and 2013 [5]. In spite of the global decline, performance on child mortality varies across countries Some countries were closer to meeting their MDG targets than others. Compared with more affluent countries, LMICs generally have higher rates of under-five mortality. Efforts in reducing U5MR in SSA led to a reduction from 180 deaths (per 1000 live births) in 1990 to 83 in 2015 [6]. However, the MDGs failed to monitor the most significant causes of child mortality; therefore, the global attention and resources weren't effectively directed towards the complex diseases contributing to child mortality. For example, measles vaccination was a single indicator for child mortality, but it was discovered that measles only accounted for 4% of U5MR [5]. Diseases such as pneumonia, diarrhoea, malaria and neonatal sepsis accounted for about 54% of U5MR in 2000 but there were no indicators directly evaluating progress in their reductions in the MDGs [5,7]. Furthermore, gains in child survival were not consistent across different regions of the world and the MDG 4 targets were not met in many African countries; hence, a new development agenda–SDG was put forward by the international community. The target 3.2 of the SDG seeks to end preventable deaths of newborns and children under-5 years of age by 2030 [8,9].

With a mortality rate of 74 deaths per 1000 live births as of 2017, the SSA region alone accounted for a large share of the global statistics on under-five deaths [1]. This figure translates to 1 child in 13 dying before their fifth birthday [10]. The disparity between SSA and the rest of the world is evident considering its child mortality figures which are 14 times higher than those of high-income countries. As the aim of SDG target 3.2 is to reduce neonatal mortality to at least 12 deaths per 1000 livebirths and under-five mortality to at least 25 deaths per 1000 live births [1,11], countries in SSA and other parts of Africa are required to double or triple their rate of reduction to meet this SDG target on child health [6].

While it is still early to assess the impacts of SDGs on child mortality rates, there are emerging concerns about the shortcomings of the SDGs to adequately address U5MR, particularly in low-resource settings [12,13]. *The Countdown Report of the 2030 Agenda for Sustainable Development* indicates that many countries in Africa lack up-to-date mortality statistics. Current data on mortality are based on extrapolations and predictions of mortality data. The WHO recommends the use of death registration as a data source; however, civil registration and vital statistics remain largely incomplete in many African countries [14,15]. Household-based surveys and censuses remain the major source that most SSA countries in particular rely on to assess trends in child mortality [13,16]. Although, with some limitations, these data sources are useful in monitoring progress and informing global decisions. For instance, a recent modelling study used these data sources and suggested that child health interventions in partnership with communities have the potential to prevent 4.9 million deaths [17].

Using a selected set of specialised surveys conducted by the World Bank, WHO, United Nations, and the DHS programme during the pre-SDG era (2010–2015) and the post-2015 development period (SDG era– 2016 to date), we explored Africa's child survival gains and its

prospects for attaining the SDG target 3.2 –reducing U5M to at least 25 deaths per 1000 live births by 2030.

## Data and method

### Methods

**Study design.**   This study undertook a multi-country secondary analysis of datasets from multiple sources. The datasets were obtained from the World Development Indicator (WDI), WHO's Global Health Observatory (GHO) Data, Demographic and Health Survey (DHS), and Human Development Report (HDR) with survey years ranging from 2006 to 2018. Similar methodologies were employed in conducting various surveys across different countries; as such, the datasets are nationally representative and comparable across countries. The surveys routinely elicit demographic, socio-economic, environmental, and health information across developing countries every 5 years. Details of data collection methodologies particularly for DHS data for different countries are available in the final reports for the selected countries.

The unit of analysis in the current study is the individual African country with the corresponding country-level estimates on children under the age of 5 years and women of reproductive ages (15–49 years). Although progress has been made in reducing under-five mortality globally, there are high rates of U5M in many parts of Africa. Besides, there are inequalities in U5MRs between and within countries. While the mortality estimates for 54 African countries were obtained from the DHS's STATcompiler and WHO's GHO data, other estimates were obtained from the WDI and HDR.

**Dependent variable.**   The dependent variable in this study was the under-five mortality rate. This was defined as the risk of a child dying before the fifth birthday. The estimates were child-based; hence the U5MR was measured in each data source and period as the total number of deaths among under-five children per 1,000 live births. The rate in each country was used as estimates for both pre-, and post-SDG years.

**Independent variables.**   Twelve (12) independent variables aggregated at the country level were used in the study. The selection of independent variables for this study was guided by the reviewed literature. We included demographic and socioeconomic factors such as the prevalence of urban residence, women's educational status, and household wealth index. Others were variables associated with maternal and child health service coverage, including antenatal care (ANC), the prevalence of skilled birth attendance, delivery in health care facilities, complete immunisation, type of birth, high-risk fertility, access to improved water, and access to sanitation. We conducted a multicollinearity test to remove highly correlated variables. Table 1 presents the operational definitions for the selected independent variables.

**Statistical analysis.**   We employed descriptive and inferential statistics in data analysis to explore the distribution of the dependent and independent variables. The values were expressed as mean with standard deviation (SD) or median with interquartile range. The SD indicates the extent of dispersion and variability among the 54 African countries. We also built statistical models to explore the core domains of the selected variables.

GeoDa v. 1.14 was used to perform the spatial analysis as it explicitly handles spatial data is suitable for this study [18]. Each country was our unit scale for spatial analysis and a distance-based threshold of 4272 km (Arc distance) was generated with X and Y coordinates from the shapefile of the African map [19]. The appropriateness of this spatial weight was confirmed by assessing the symmetry of the connectivity histogram, and all 54 countries were interlinked; a necessary feature to ascertain spatial dependency. We generated a quantile cluster map to show the descriptive distribution of the U5MR in Africa.

**Table 1. Independent variables for modelling child survival determinants in Africa.**

| Variables | Operational definitions/variable categories |
|---|---|
| Secondary/higher Education | Proportion of the women of childbearing age with secondary/higher education |
| Urban residence | Proportion of the women of childbearing age resident in urban areas |
| Middle/higher wealth quintile | Proportion of women living in households with a minimum of middle wealth quintile |
| ANC visit | Proportion of women of childbearing age with four or more antenatal care visit |
| Skilled birth attendance | Proportion of child deliveries undertaken by health professional(s) such as a doctor, nurse, midwife, trained health worker or trained birth attendants |
| Delivery in healthcare facility | Proportion of child deliveries occurring in health facilities |
| High-risk fertility | Proportion of women of childbearing age with one or more high-risk births categorised as normal (no high-risk birth), too early (birth before age 18), too many (more than 4 births), too late (births at age 35 or above), too close (successive births with less than 33-month interval) |
| Contraceptive use | Proportion with current use of any modern contraceptive method |
| Type of birth | Type of birth at a given point of delivery categorised as single birth and multiple birth (for twins or triplet or more) |
| Completed Immunisation | Proportion of under-five children with completed immunisation |
| Access improved water | Proportion living in households with access to water from improved sources |
| Improved sanitation | Proportion living in households with improved sanitation |

Global Moran's I analysis was first conducted to examine if spatial autocorrelation occurs at the local level, after which Local indicators of spatial autocorrection (LISA) cluster maps were generated to statistically show the hot and cold spots spatial clusters of neighbouring countries with high and low U5MR. Local spatial autocorrelation was measured with Local Moran's I index which ranges from -1 to +1. Positive values indicate strong clustering while negative values indicate dispersion. Finally, we conducted spatial regression based on approaches developed by Anselin [20] to investigate the association between U5M and the independent variables. Three models were built to reflect different domains of the variables and the final model contained only significant variables from each model. A pair-wise correlation was used to deal with missing data which was less than 2%. Ordinary least square (OLS) diagnostic was examined for each model, where spatial dependency was indicated, the model was fitted with spatial error or spatial lag regression as appropriate; the best-fit model was determined using R-squared, Log-likelihood, and Akaike information criterion. A multicollinearity test value of <30 was used to select variables in each model, and finally, 999 Monte Carlo permutation was used for randomisation to ensure a p-value <0.05.

Bayesian network analysis was further used to examine the strength of the relationship between U5MR and the independent variables that were found to be significant in the spatial regression models. The structure of the network was built using score-based structure algorithms and the temporal precedence of the variables from the domain knowledge of the authors. Geographical regions, based on the United Nation's categorisation, were introduced as a variable to deal with possible residual confounding. First, we explored the diagnostic conditional probability distribution after we discretised the continuous variables, and then analysed them to show the strength of association between the independent variables and U5MR. Finally, supervised machine learning techniques were employed to predict the probability of significant independent variables from the Bayesian network when countries with higher U5MR have reductions to 25 deaths per 1000 live births–the target for the 2030 SDGs. K-fold

cross-validation (at k = 10) was employed to compare and examine the model's goodness of fit; log-likelihood loss was used as the loss function, therefore, the lower the value, the better the fit [21]. Bayesian network analysis was conducted with R version 4.0 using the "bnlearn" package majorly.

**Ethics statement.** This study involved the analysis of secondary datasets from the Demographic and Health Survey

Program, the World Development Indicator, WHO's Global Health Observatory Data, and Human Development Report (HDR). All the data are aggregate-level indicators, with no unique identifiers, except the DHS datasets which emanated from surveys. To conduct the DHS surveys, its protocol was approved by the ICF Institutional Review Board and ethics committees of the selected countries. Although the study focused on children below age 5, the study participants were women of reproductive ages (15–49 years) who responded to the questions on their birth histories. Permission to use the DHS data for this study was obtained from the ICF International. The datasets obtained for analysis were anonymised and all unique identifiers were removed before the release of the data for public use.

## Results

The results in Table 2 indicate changes in the trends of infant mortality rate (IMR) and U5MR in Africa, especially in the SSA region. While the rates remain high with minimal changes in some SSA countries, others have achieved significant decline within 4-year to 7-year intervals. Prior to the introduction of SDGs, the highest IMR, measured per 1,000 livebirths, were recorded in Central African Republic (95.6), Sierra Leone (95.5), Somalia (88.0), Chad (79.8), DRC (77.9), Lesotho (73.0) and Equatorial Guinea (72.4). The least were recorded in Libya (12.4), Seychelles (12.5), Mauritius (12.8), Tunisia (15.0) and Cabo Verde (19.5). By 2018, none of the countries with the highest IMR has achieved any significant decline. For instance, none of the countries with 50 or more infant deaths (per 1,000 live births) has achieved up to 20% decline in IMR except Malawi with a 36.4% decline from 66.0 to 42.0 deaths; Cameroon with 22.6% decline from 62.0 to 48.0; Uganda and Swaziland with 20.4% decline each from 54.0 to 43.0 deaths. The lowest percentage decline was recorded in South Sudan (0.5%), Nigeria (2.9% decline from 69.0 to 67.0), Mali (3.6% decline from 56.0 to 54.0) and Guinea with no changes. Most of the other African countries with less than 50 infant deaths per 1,000 live births have achieved about 10.3–21.7% decline, except Zambia (6.7% decline), Benin with 31.0% increase and Mauritius (6.2% increase).

Of the 54 African countries under study, only eight countries: Libya, Tunisia, Egypt, Morocco and Algeria from North Africa, Seychelles and Mauritius from East Africa and Cabo Verde from West Africa have achieved U5MR reduction to a maximum of 25 deaths per 1,000 live births. While Sao Tome Principe, South Africa, Rwanda, Botswana, Namibia and a few other East African countries are close to the target, all other African countries are far from the goal.

Examining the trends, the U5MR was more than 100 deaths per 1,000 livebirths (pre-SDG era) in 11 countries: Somalia (142.3), Central African Republic (138.0), Chad (137.9), Sierra Leone (136.7), Nigeria (128.0), Guinea (123.0), Cameroon (122.0), Malawi (112.0), DRC (104.2), Niger (103.2) and Equatorial Guinea (100.7). By 2018, while Malawi and Cameroon had recorded 42.9% and 34.4% decline, respectively, others had recorded about 13.7–23.1% decline except Guinea (9.8% decline) and Nigeria which had conversely recorded about 3.1% increase. Among countries with below 100 under-five deaths per 1,000 live births, Morocco, Burkina Faso, Ghana, Senegal, Swaziland, Ethiopia, Rwanda, Angola, Zimbabwe and Uganda have achieved a substantial decline of about 20.0–28.9%, while others have cut under-five

**Table 2. Gains/Losses in infant and under-five mortality rates in Africa from year 2013 to 2018.**

| Country | Pre-SDG Infant Mortality Rate (2013) | Infant Mortality Rate During SDG (2018) | Percentage change in infant mortality | Pre-SDG Under-5 Mortality Rate (2013) | Under-5 Mortality Rate During SDG (2018) | Percentage change in under-5 mortality |
|---|---|---|---|---|---|---|
| **North Africa** | | | | | | |
| **Algeria** | 22.1 | 20.1 | 9.0 | 25.7 | 23.5 | 8.6 |
| **Egypt** | 21.5 | 18.1 | 15.8 | 25.5 | 21.2 | 16.9 |
| **Libya** | 12.4 | 10.2 | 17.7 | 14.5 | 12.0 | 17.2 |
| **Morocco** | 24.0 | 19.2 | 20.0 | 28.0 | 22.4 | 20.0 |
| **Sudan** | 47.6 | 42.1 | 11.6 | 70.1 | 60.5 | 13.7 |
| **Tunisia** | 15.0 | 14.6 | 2.7 | 17.5 | 17.0 | 2.9 |
| **Central Africa** | | | | | | |
| **Angola (2011, 2016)\*** | 50.0 | 44.0 | 12.0 | 91.0 | 68.0 | 25.3 |
| **Cameroon (2011, 2018)\*** | 62.0 | 48.0 | 22.6 | 122.0 | 80.0 | 34.4 |
| **CAR** | 95.6 | 84.5 | 11.6 | 138.0 | 116.5 | 15.6 |
| **Chad** | 79.8 | 71.4 | 10.5 | 137.9 | 119.0 | 13.7 |
| **Congo** | 40.7 | 36.2 | 11.1 | 57.9 | 50.1 | 13.5 |
| **DRC** | 77.9 | 68.2 | 12.5 | 104.2 | 88.1 | 15.5 |
| **Equatorial Guinea** | 72.4 | 62.6 | 13.5 | 100.7 | 85.3 | 15.3 |
| **Gabon** | 38.3 | 32.7 | 14.6 | 55.7 | 44.8 | 19.6 |
| **Sao Tome and Principe** | 29.2 | 24.4 | 16.4 | 38.4 | 31.2 | 18.8 |
| **East Africa** | | | | | | |
| **Burundi (2010, 2017)\*** | 59.0 | 47.0 | 20.3 | 96.0 | 78.0 | 18.8 |
| **Comoros** | 58.9 | 51.3 | 12.9 | 79.5 | 67.5 | 15.1 |
| **Djibouti** | 57.4 | 49.8 | 13.2 | 69.6 | 59.3 | 14.8 |
| **Eritrea** | 36.0 | 31.3 | 13.1 | 49.8 | 41.9 | 15.9 |
| **Ethiopia (2011, 2016)\*** | 59.0 | 48.0 | 18.6 | 88.0 | 67.0 | 23.9 |
| **Kenya** | 35.8 | 30.6 | 14.5 | 50.1 | 41.1 | 18.0 |
| **Madagascar** | 42.6 | 38.2 | 10.3 | 61.2 | 53.6 | 12.4 |
| **Malawi (2010, 2016)\*** | 66.0 | 42.0 | 36.4 | 112.0 | 64.0 | 42.9 |
| **Mauritius** | 12.8 | 13.6 | +6.2 | 14.6 | 15.5 | +6.2 |
| **Mozambique** | 63.1 | 54.0 | 14.4 | 89.9 | 73.2 | 18.6 |
| **Rwanda** | 34.5 | 27.0 | 21.7 | 47.5 | 35.3 | 25.7 |
| **Seychelles** | 12.5 | 12.4 | 0.8 | 14.5 | 14.5 | 0.0 |
| **Somalia** | 88.0 | 76.6 | 13.0 | 142.3 | 121.5 | 14.6 |
| **South Sudan** | 64.0 | 63.7 | 0.5 | 99.1 | 98.6 | 0.5 |
| **Tanzania (2010, 2016)\*** | 51.0 | 43.0 | 15.7 | 81.0 | 67.0 | 17.3 |
| **Uganda (2011, 2016)\*** | 54.0 | 43.0 | 20.4 | 90.0 | 64.0 | 28.9 |
| **Zambia (2014, 2018)\*** | 45.0 | 42.0 | 6.7 | 75.0 | 61.0 | 18.7 |
| **Zimbabwe** | 42.8 | 33.9 | 20.8 | 62.3 | 46.2 | 25.8 |
| **West Africa** | | | | | | |

*(Continued)*

**Table 2.** (Continued)

| Country | Pre-SDG Infant Mortality Rate (2013) | Infant Mortality Rate During SDG (2018) | Percentage change in infant mortality | Pre-SDG Under-5 Mortality Rate (2013) | Under-5 Mortality Rate During SDG (2018) | Percentage change in under-5 mortality |
|---|---|---|---|---|---|---|
| Benin (2012, 2018)* | 42.0 | 55.0 | +31.0 | 70.0 | 96.0 | +37.1 |
| Burkina Faso | 57.5 | 49.0 | 14.8 | 95.8 | 76.4 | 20.3 |
| Cabo Verde | 19.5 | 16.7 | 14.4 | 22.9 | 19.5 | 14.8 |
| Cote d'Ivoire | 68.9 | 59.4 | 13.8 | 96.7 | 80.9 | 16.3 |
| Gambia | 43.9 | 39.0 | 11.2 | 69.3 | 58.4 | 15.7 |
| Ghana | 42.0 | 34.9 | 16.9 | 60.2 | 47.9 | 20.4 |
| Guinea (2012, 2018)* | 67.0 | 67.0 | 0.0 | 123.0 | 111.0 | 9.8 |
| Guinea-Bissau | 63.8 | 54.0 | 15.4 | 98.7 | 81.5 | 17.4 |
| Liberia | 62.5 | 53.5 | 14.4 | 85.1 | 70.9 | 16.7 |
| Mali | 56.0 | 54.0 | 3.6 | 95.0 | 101.0 | +6.3 |
| Mauritania | 58.1 | 51.5 | 11.4 | 89.0 | 75.7 | 14.9 |
| Niger | 55.2 | 48.0 | 13.0 | 103.2 | 83.7 | 18.9 |
| Nigeria | 69.0 | 67.0 | 2.9 | 128.0 | 132.0 | +3.1 |
| Senegal | 43.0 | 37.0 | 14.0 | 65.0 | 51.0 | 21.5 |
| Sierra Leone | 95.5 | 78.5 | 17.8 | 136.7 | 105.1 | 23.1 |
| Togo | 54.2 | 47.4 | 12.5 | 81.8 | 69.8 | 14.7 |
| Southern Africa | | | | | | |
| Botswana | 34.8 | 30.0 | 13.8 | 43.2 | 36.5 | 15.5 |
| Eswatini/ Swaziland | 54.0 | 43.0 | 20.4 | 69.4 | 54.4 | 21.6 |
| Lesotho | 73.0 | 65.7 | 10.0 | 94.7 | 81.1 | 14.4 |
| Namibia | 35.3 | 29.0 | 17.8 | 46.6 | 39.6 | 15.0 |
| South Africa | 34.1 | 28.5 | 16.4 | 41.2 | 33.8 | 18.0 |

SDG: Sustainable Development Goal

*The pre- and post-SDG survey years used

+ increase.

mortality by 12.4–19.6%. However, South Sudan, Algeria, Tunisia and Seychelles recorded a less than 10% decline while other countries (Mali, Benin, and Mauritius recorded an increase.

The values in Tables 2 and 3 alongside the visual representation in Fig 1 showed the distribution of U5MR across the 54 African countries. As shown in Table 3, the average U5MR is 61.8 (SD = 55.9) per 1, 000 live births, with the rates ranging from 11.9 deaths in Libya to 132.0 per 1,000 live births in Nigeria. The West African sub-region has the highest U5MR, with an average of 78.4 deaths (per 1, 000 live births)–ranging from 19.5 in Cape Verde to 132.0 in Nigeria. On the other hand, Northern African countries had the lowest U5MR with an average of 36.1 deaths (per 1,000 live births), with the rates ranging from 11.9 in Libya to 60.5 in Sudan. Southern and Middle African countries, respectively had an average of 52.4 and 75.8 U5MR per 1,000 live births.

The global Moran's I statistic revealed a statistically positive spatial autocorrelation with Moran's I index 0.13 (p<0.05), therefore, suggesting the aggregation of neighbourhood countries with similar values of U5MR across geographical space than an expected random distribution. As shown in Fig 2, the LISA map generated further showed the presence of high-high clusters (countries with significantly high U5MR forming hot spots) in 7 (13%) of the 54

**Table 3. Descriptive analysis of U5MR and selected independent variables (SD- Standard Deviation; IQR- Interquartile range).**

| Indicators | Mean | Median | IQR | SD | Min | Max |
|---|---|---|---|---|---|---|
| ANC visit | 57.7 | 57.2 | 24.7 | 17.9 | 6.3 | 89.3 |
| Access to improved water | 69.9 | 69 | 22.8 | 16.5 | 39.0 | 100.0 |
| Delivery in healthcare facility | 71.1 | 80.8 | 29.1 | 23.2 | 9.4 | 100.0 |
| Contraception | 50.9 | 47.7 | 33.4 | 20.3 | 17.5 | 86.6 |
| Completed Immunization | 58.4 | 64.6 | 36.8 | 23.8 | 7.35 | 92.7 |
| High risk fertility | 42.4 | 42.5 | 14.0 | 9.9 | 22.6 | 60.6 |
| Improved sanitation | 42.9 | 37.5 | 39.5 | 26.9 | 7.0 | 100.0 |
| Secondary/higher Education | 30.7 | 28.0 | 26.7 | 21.4 | 1.7 | 88.8 |
| Skilled birth attendance | 72.2 | 78.0 | 26.5 | 23.0 | 9.0 | 100 |
| Under-five mortality rate | 61.8 | 55.9 | 41.4 | 29.9 | 11.9 | 120.3 |
| Urban residence | 45.4 | 43.5 | 23.9 | 17.8 | 13.6 | 88.0 |
| Wealthy | 57.8 | 58.1 | 2.7 | 2.2 | 52.4 | 63.5 |

countries. The hotspots for U5MR in Africa were found in Western (Nigeria, Niger, Togo, Burkina Faso, and Middle (Cameroon, Chad and Central Africa Republic) regions of the continent. On the other hand, the cold-spot cluster was observed in Tunisia only. Outside these two extremes, four countries (Gabon, Congo, Ghana and Cote d'Ivoire) formed low-high clusters. This indicates that these countries have significantly low U5MR within the neighbourhood of countries with significantly high U5MR.

Table 4 presents findings from the spatial regression analysis. In model 1 that adjusted for socioeconomic factors, the prevalence of secondary school education completion by females (-0.49) and access to water (-0.79) were significantly associated with U5MR at $p<0.05$. However, when maternal and child health service indicators were adjusted for in Model 2, only the prevalence of contraception (-0.67) was found to be negatively associated with risk of U5MR at $p<0.05$.

In the fully adjusted model where only the significant variables from Models I and II were considered, the prevalence of contraceptives was the only significant determinant (-0.491) at $p<0.05$. Although the prevalence of access to water was not significant in the final model, it had the strongest association with U5MR when socioeconomic data alone were adjusted for in previous models. Extrapolation from the final regression model indicates that every 1% increase in the prevalence of contraceptives across African countries reduces the spatial prevalence of U5MR by 51% (adjusted $R^2 = 0.35$). The adjusted LISA maps in Fig 4 corroborate findings from the spatial regression.

Fig 3 presents the joint probability distribution from the network developed based on expert and previous research findings. It shows the local distribution corresponding to each variable in the network. Each arch (arrow) from one variable to another means the latter (variable at the head of the arrow) is dependent on the former (variable at the tail end of the arrow). The region is the only variable with no parent, meaning it is not dependent on any other variable while U5MR is dependent on other variables. However, Fig 4 further showed that among the independent variables included in the network, the prevalence of contraception has the strongest influence on U5MR as indicated by the width of the arc. Interestingly, access to water and completion of secondary school by females also showed a direct influence on U5MR. It is also important to mention that region as a variable only had a significant influence on the prevalence of contraception, highlighting its indirect influence on U5MR since contraception had the strongest and most significant influence. In the network, we also found that the prevalence of access to water showed a significant relationship with sanitation and the

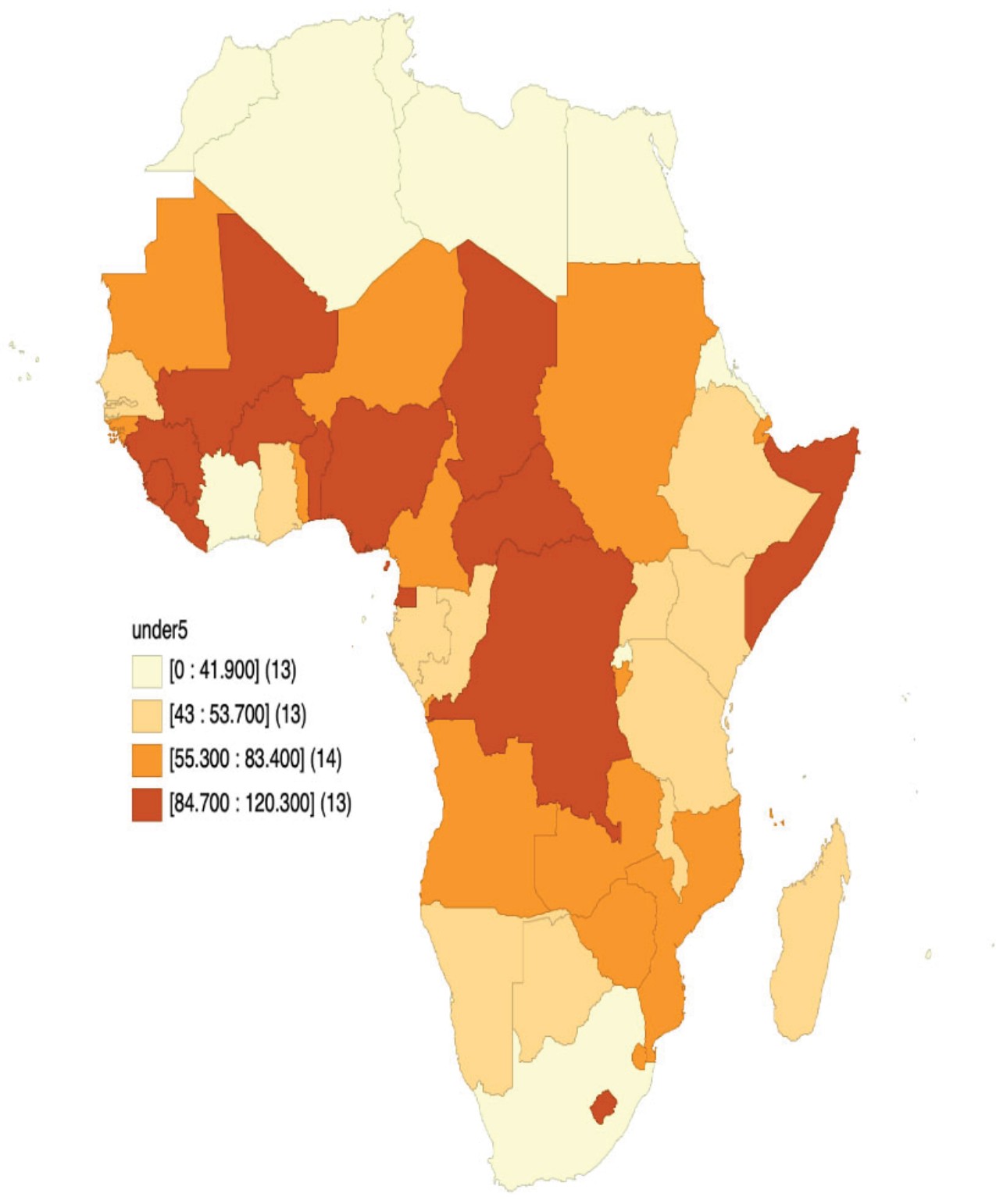

under5

[0 : 41.900] (13)

[43 : 53.700] (13)

[55.300 : 83.400] (14)

[84.700 : 120.300] (13)

**Fig 1. Distribution of U5MR across 54 African countries.**

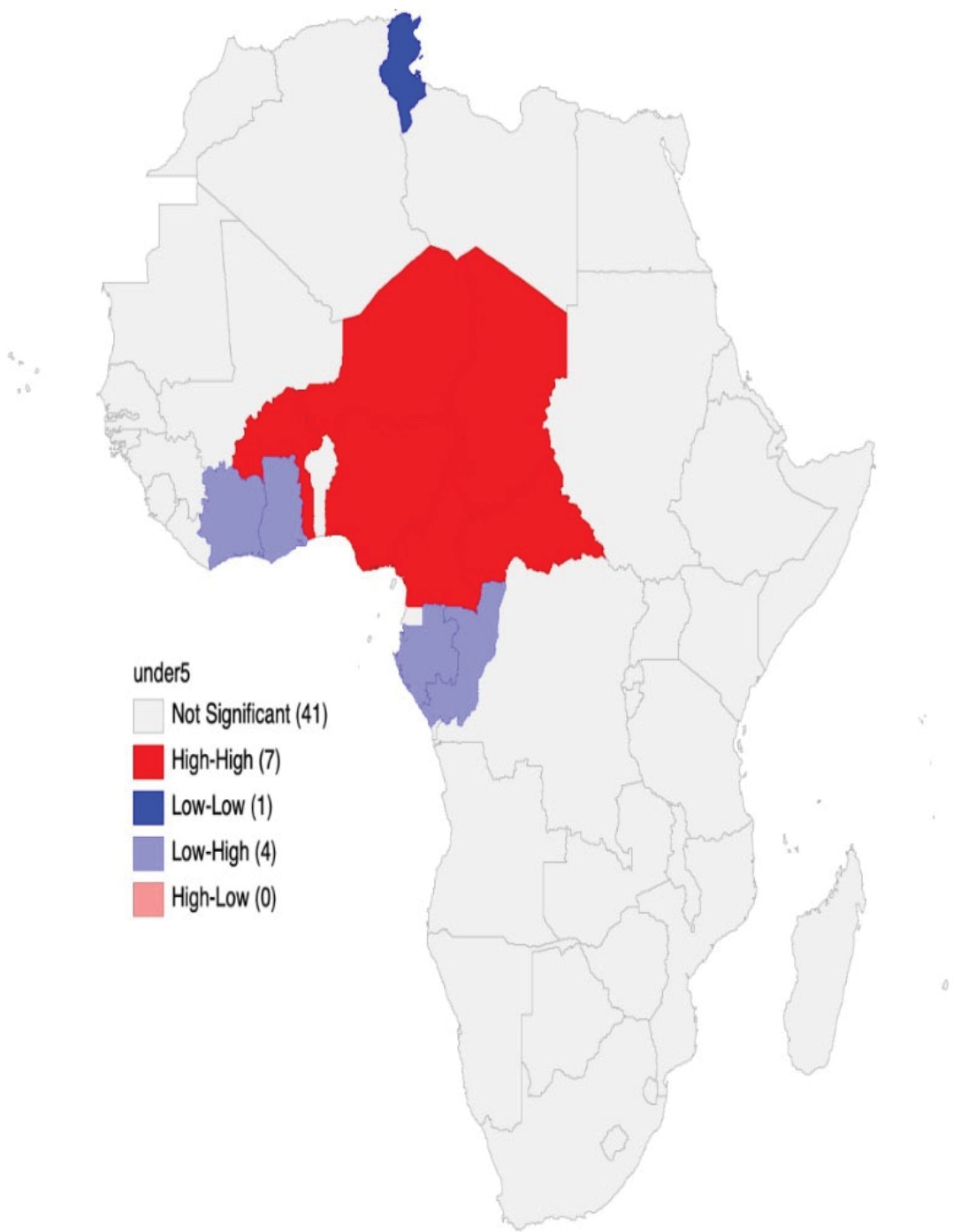

**Fig 2. Local indicator of spatial autocorrelation cluster map showing the hot and cold spots of U5MR across 54 African countries.**

prevalence of skilled birth attendance was strongly related to the prevalence of institutional birth.

Finally, as presented in Table 5, results of the predictions, using supervised machine learning on the Bayesian network, revealed that the joint probability for low U5MR (25 deaths per 1000 live birth) increases from 21.6% to 100% when the contraceptive prevalence increases

**Table 4. Multivariable spatial regression model for under-five mortality rate.**

| Variable | Model 1 β (p-value) | Model 2 β (p-value) | Model 3 β (p-value) |
|---|---|---|---|
| **Socioeconomic/Culture** | | | |
| Urban Residence | 0.29 (0.31) | | - |
| Female secondary school education status | -0.49(0.25) | | -0.35 (0.2) |
| Access to basic water | -0.79 (0.36) | | 0.44 (0.26) |
| **Service coverage** | | | |
| Contraceptive | | -0.67 (0.28) | -0.49 (0.03) |
| Skilled birth attendance | | -0.51(0.27) | 0.21 (0.41) |
| ANC coverage—at least four visits | | -0.05 (0.33) | 0.44 (0.26) |
| High fertility risk | | 0.32 (0.59) | - |
| **Adjusted R2** | 0.28 | 0.25 | 0.35 |
| **Model Estimation** | OLS | OLS | OLS |

from 49.8% to 78.5% and skilled birth attendance increases from 44.8% to 86.3% and percentage of secondary school completion of females increases from 42.5 to 74.0%. The test of goodness of fit showed that the log-likelihood loss was 5.6 and classification error (0.09) from the cross-validation analysis showed some levels of good fit and prediction accuracy of the Bayesian model built, showing about 90% accuracy.

## Discussion

Global policy and public health stakeholders acknowledge child survival outcomes as an important socioeconomic and healthcare indicator for monitoring progress on relevant SDGs. This study explored Africa's child survival gains and prospects of meeting the SDG target 3.2 (reducing U5MR to at least 25 deaths per 1000 live births by 2030). The study also examined the factors associated with child mortality in Africa. Despite significant global declines in U5M over the past two decades [22], this study observed that only eight (Libya, Tunisia, Egypt, Morocco, Algeria, Seychelles, Mauritius, and Cabo Verde) out of 54 African countries had achieved the SDG 3.2 target and none of the countries with the highest infant mortality rates (such as Sierra Leone, Central Republic Africa, Democratic Republic of Congo, Chad and Nigeria) had achieved any significant decline.

This study observed an average U5MR of 61.8 per 1000 live births, with huge disparities across regions and between countries in Africa. For instance, U5MR ranged from 11.9 per 1000 live births in Libya to 120.3 per 1000 live births in Somalia. Similarly, the North African region had the lowest U5MR average (36.1 deaths per 1000 live births) when compared to Southern Africa (52.4), Middle Africa (75.8), and West Africa with the highest average of 78.4 deaths per 1000 live births. In this study, the disparities in U5MR observed between SSA and North Africa largely reflect differences in shared historical, cultural, public health, and socioeconomic characteristics between and across African countries [23]. Using DHS data, Yaya et al. [24] adduced socioeconomic, demographic, and health system factors as reasons for SSA's high U5MR and poor child survival outcomes.

Poor child survival gains observed among SSA countries with little progress towards the SDG target 3.2 demonstrate the need to accelerate progress through concerted approaches tailored to address each country's context and child survival priorities [22,24]. In contrast, North African countries such as Egypt, Morocco, and Tunisia have received credits for meeting the SDG targets on reducing under-five mortality through increased access to neonatal and obstetric care, quality improvements in antenatal and delivery services, decisive leadership for

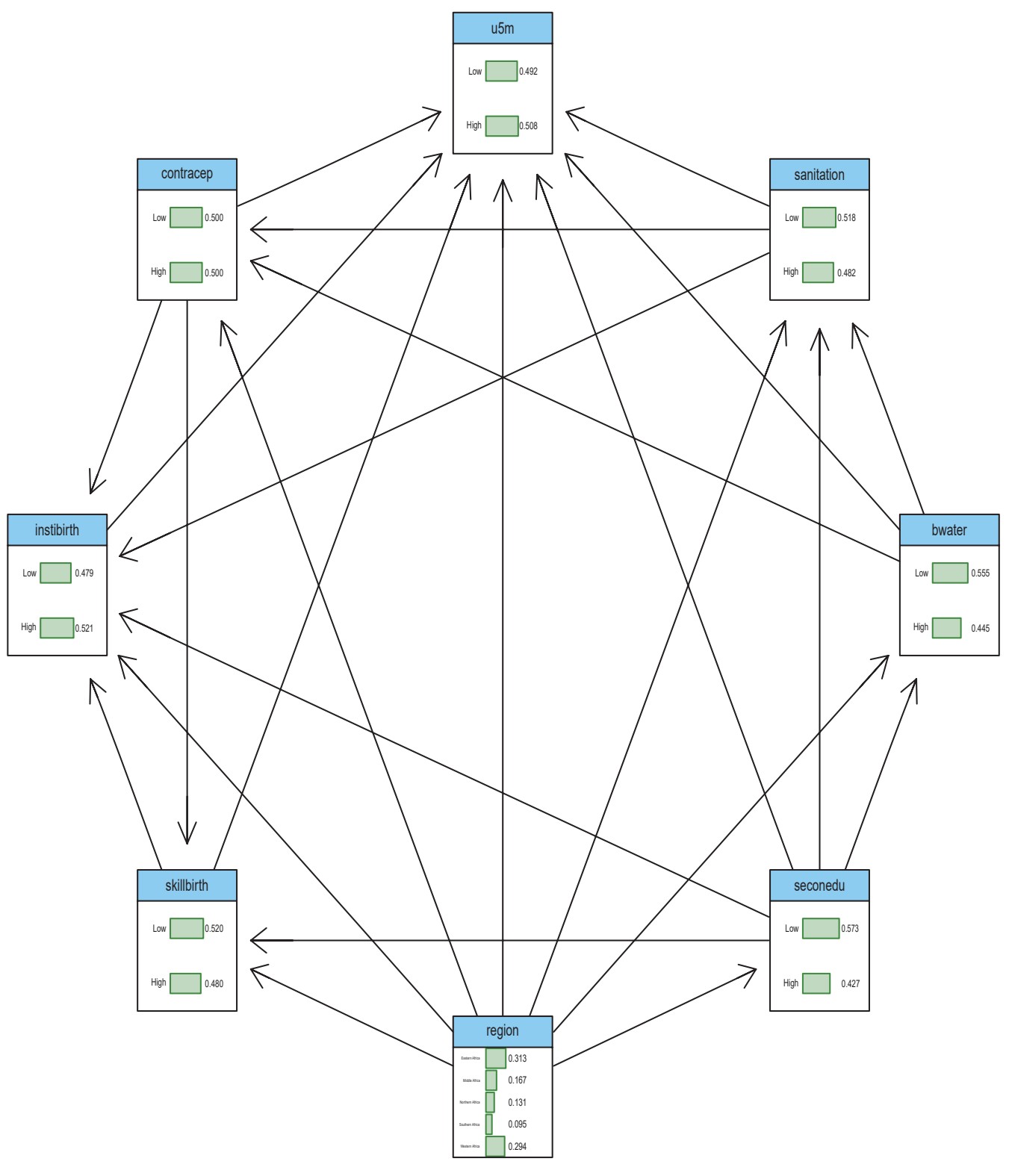

**Fig 3. Spatial autocorrelation from Local Moran's I index.**

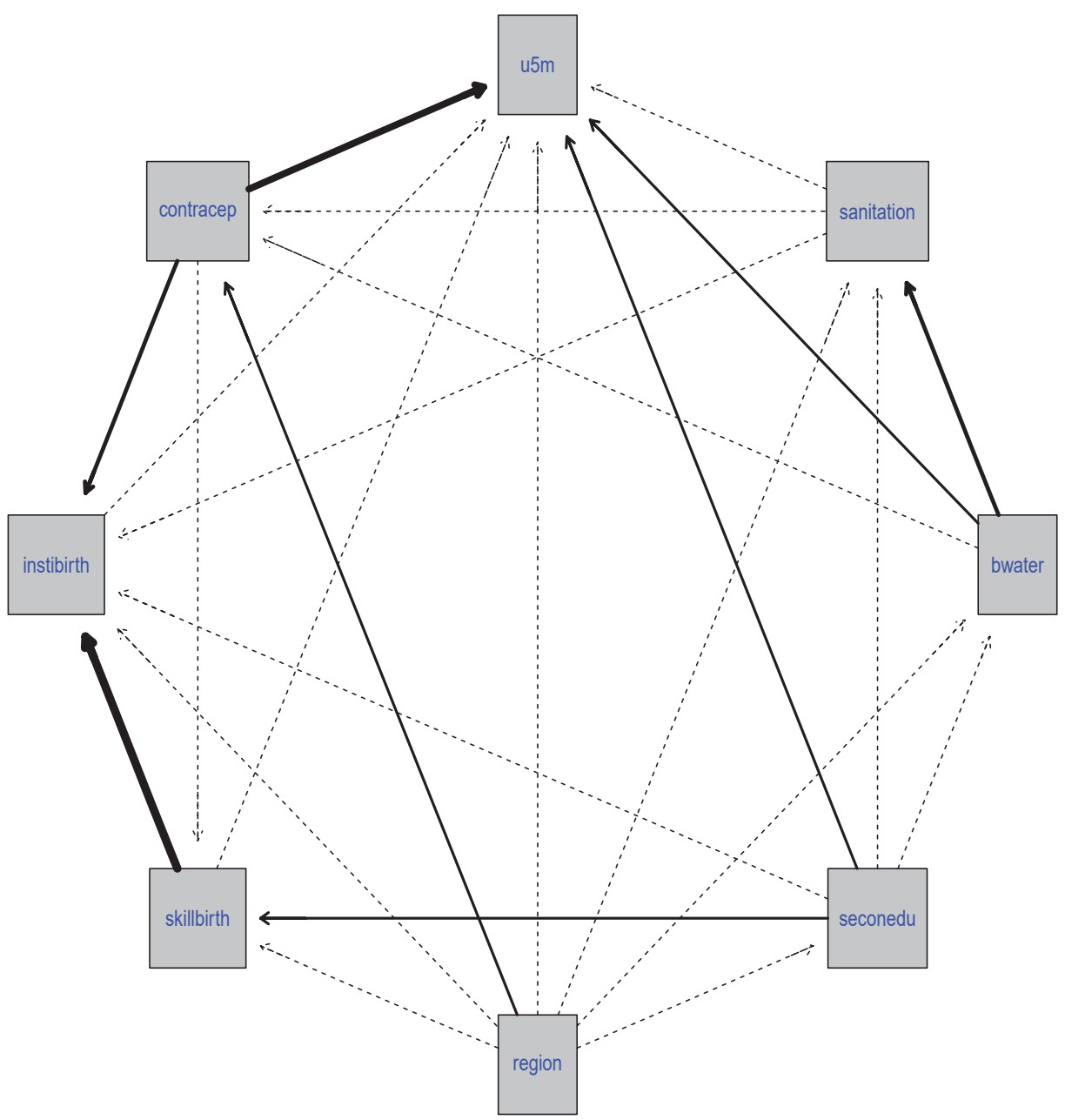

**Fig 4. Adjusted Local indicator of spatial autocorrelation map.**

maternal and child health programmes, and extensive implementation of the Integrated Management of Childhood Illness (IMCI) strategy [23,25,26]. The significant child survival gains of a considerably large number of Northern African countries are pointers that reduction of U5MR is possible in SSA through strategic socioeconomic, public health, and political commitments and prioritisation of programmes on child survival.

Study findings revealed a statistically significant Moran's I index (Fig 3), which indicates a clustering of African countries with similar U5MR across geographical space. It is concerning that seven of the countries (Nigeria, Niger, Togo, Burkina Faso, Cameroon, Chad, and the

**Table 5. Inferential joint probability of U5MR, prevalence of contraception, and skill birth attendant and female completion of secondary school education from the Bayesian Network.**

| Variables | Baseline (%) | Prediction (%) |
|---|---|---|
| U5MR (Low) | 21.6 | 100 |
| Contraception (High) | 49.8 | 78.5 |
| Skill birth attendant (High) | 44.8 | 86.3 |
| Female completion of secondary school education (High) | 42.5 | 74.0 |

U5MR = Under-five mortality rate.

Central Africa Republic) that formed high-high clusters (hot spots of countries with significantly high U5MR) were in the SSA region. Interestingly, only Tunisia formed a cold-spot cluster, while four countries (Gabon, Congo, Ghana and Cote d'Ivoire) were in the low-high cluster due to their significantly low U5MR within the neighbourhood of countries with considerably high U5MR. Variations in the introduction and extent of implementation of under-five focused policies and interventions, socioeconomic determinants and prevalent mortality causes are plausible reasons for the structuring of African countries into high- and low-cluster areas in this study [27,28].

The prevalence of predisposing factors such as poor access to quality health care services, poverty, female illiteracy, limited access to safe water, or diseases such as malaria, pneumonia, and diarrhoea, which are leading U5M causes, may account for high-high clusters formed by seven SSA countries [24,29–31]. Conversely, the elimination of malaria–which is a leading cause of U5M, high child immunisation coverage rates, achievement of less than 5% undernourished individuals between 1990 and 2015, and Tunisia's high levels of improved water and sanitation could be possible reasons for the country's formation of a cold-spot in Africa [32]. By identifying neighbourhood countries with similar U5MR outcomes, this study provides critical evidence for health planning and allocation of resources to neighbourhoods with high U5MR in Africa.

Bivariate spatial regression and Bayesian network analysis were employed to identify the factors associated with child mortality in Africa. Upon adjustment for socioeconomic characteristics, results from the first bivariate spatial regression model highlighted the prevalence of female secondary school education completion and access to water as statistically significant predictors of U5MR in Africa. The prevalence of contraception maintained a significant association with U5MR after adjusting for maternal and child health service indicators and in the final model where significant variables were adjusted. Interestingly, findings from the Bayesian network analysis further reinforced the results of the bivariate spatial regression as access to water, completion of secondary school, and prevalence of contraception were associated with U5MR, with the prevalence of contraception having the strongest influence.

Furthermore, the Bayesian network prediction of this study shows an increase in the joint probability of low U5MR (25 deaths per 1000 live births) from 21.6% to 100% when the percentage of female secondary school completion increases from 42.5 to 74.0%. This finding highlights the need for greater investments in female education as a critical intervention to improve child health outcomes, particularly in SSA.

Both the bivariate spatial regression and Bayesian network analysis identified the prevalence of contraception as the most significant variable associated with U5MR. As has been previously established [29,33,34], this finding indicates the implication of low contraceptive uptake for high-risk births and increased child mortality in SSA. Effective contraceptive use allows women to lengthen birth intervals, limit the number of births, achieve their fertility desires,

and space children [35]. Beyond opportunities to space and limit the number of children, contraceptive use contributes to under-five mortality decline by averting high-risk pregnancies at extremes of maternal age, i.e. too young or too old, and any unintended pregnancies due to the mother's physical and socioeconomic unreadiness for childbirth [36–38]. The study predicts that increasing contraceptive prevalence from 49.8% to 78.5% will facilitate the achievement of low U5MR (25 deaths per 1000 live births). This finding suggests the need for African countries with high U5MR to address the problem of poor access to family planning and reproductive health services in order to accelerate progress in child health and survival.

## Conclusion

This study provides an important insight into Africa's child survival gains and the prospects for meeting SDG target 3.2 (reducing U5MR to at least 25 deaths per 1000 live births by 2030). This study revealed that eight countries mainly from North Africa, have achieved the SDG target 3.2. While a few are close to the target, most SSA countries are far from meeting the target. As 2030 approaches, the SSA countries that formed the high-high clusters (Nigeria, Niger, Togo, Burkina Faso, Cameroon, Chad, and the Central Africa Republic) may not achieve the SDG target 3.2 except urgent and right investments are made. There is a need to address poor access to quality health care, poverty, female illiteracy, limited access to safe water, and poor access to quality family planning services. Considering the insufficient progress towards achieving SDG target 3.2 in SSA, a much stronger framing, better coverage and more effective child health interventions are urgently needed to accelerate child survival gains in the region. The results of this study have utility for informing and guiding programmatic interventions towards improving child health and survival in SSA and Africa as a whole.

## Study strength and limitations

The use of multiple large datasets has its strengths as it permits comparability of findings across settings and countries, albeit, secondary data has some drawback. First, it restricts analysis only to the available information already collected during the survey. Second, the study design is cross-sectional, thus only association can be inferred, while causality cannot be implied. Also, while the choice of the SDG target 3.2 is underpinned by measurability and relevance, some scholars are critical and sceptical about its global applicability as reducing under-five mortality to at least 25 deaths per 1000 livebirths is perceived to be overly ambitious and perhaps lacks sufficient reflection of national priorities for LMICs. Caution is advised when interpreting and rating a country's child survival progress as countries are encouraged to define their strategies to reduce U5MR in the context of national realities.

## Acknowledgments

The authors would like to thank the ICF International and the Statistical Agencies of the selected countries for permission to use the DHS data.

## Author Contributions

**Conceptualization:** Sunday A. Adedini.

**Data curation:** Sunday A. Adedini, Seun Stephen Anjorin.

**Formal analysis:** Sunday A. Adedini, Seun Stephen Anjorin, Jacob Wale Mobolaji.

**Methodology:** Sunday A. Adedini, Seun Stephen Anjorin, Jacob Wale Mobolaji.

**Supervision:** Sunday A. Adedini, Sanni Yaya.

**Writing – original draft:** Sunday A. Adedini, Seun Stephen Anjorin, Jacob Wale Mobolaji, Elvis Anyaehiechukwu Okolie, Sanni Yaya.

**Writing – review & editing:** Sunday A. Adedini, Seun Stephen Anjorin, Jacob Wale Mobolaji, Elvis Anyaehiechukwu Okolie, Sanni Yaya.

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
