## [Decision Letter · Decision Letter 0]

3 Jan 2024

PGPH-D-23-02289

Assessing Africa’s child survival gains and prospects for attaining SDG target on child mortality

Dear Dr. Adedini,

Thank you for submitting your manuscript to PLOS Global Public Health. After careful consideration, we feel that it has merit but does not fully meet PLOS Global Public Health’s publication criteria as it currently stands. Therefore, we invite you to submit a revised version of the manuscript that addresses the points raised during the review process.

It is an interesting study where authors investigated Africa’s child survival gains and prospects for attaining SDG target on child mortality. However, some technical issues should be addressed further, especially english language as mentioned by the reviewers. For example, "Results of the predictions using supervised machining learning on the Bayesian network 28 reveals that the joint probability for low U5MR (25 death per 1000 live birth) increases from 29 21.6% to 100% when the contraceptive prevalence increases from 49.8% to 78.5% and the use of 30 skilled birth attendants increases from 44.8% to 86.3% and percentage of secondary school 31 completion of female increases from 42.5 to 74.0%." in the abstract is another example, which cannot be comprehended clearly. Authors are suggested to improve manuscript by modifying English expression as possible as they can.

We look forward to receiving your revised manuscript.

Kind regards,

Shaonong Dang, PhD

Academic Editor

Journal Requirements:

Additional Editor Comments (if provided):

Reviewers' comments:

Reviewer's Responses to Questions

**Comments to the Author**

1. Does this manuscript meet PLOS Global Public Health’s publication criteria? Is the manuscript technically sound, and do the data support the conclusions? The manuscript must describe methodologically and ethically rigorous research with conclusions that are appropriately drawn based on the data presented.

Reviewer #1: Yes

Reviewer #2: Partly

2. Has the statistical analysis been performed appropriately and rigorously?

Reviewer #1: Yes

Reviewer #2: Yes

3. Have the authors made all data underlying the findings in their manuscript fully available (please refer to the Data Availability Statement at the start of the manuscript PDF file)?

Reviewer #1: Yes

Reviewer #2: Yes

4. Is the manuscript presented in an intelligible fashion and written in standard English?

Reviewer #1: Yes

Reviewer #2: No

5. Review Comments to the Author

Reviewer #1: I have read the referred article with keen interest. The information is interesting and innovative; conclusion section is interesting and authors can improve it further. I am recommending authors to do a little more work and add latest literate to support the study. The authors need to improve results section. The level of English is good and smooth, e.g., the language standard, specifically the grammar, of sufficient quality to meet scientific merit for publication. However, I suggest authors to double check for language quality. Describe scientific contribution of the study to the existing body of knowledge. I endorse this manuscript after minor revision as suggested. The topic is interesting and worthy of attention. The methodology is adequate and the conclusions are consistent with the reported data. The manuscript can be improved by expanding the references and citing some recently published articles on this topic.

Authors should consider the following recommendations:

Shuja, K. H., Aqeel, M., & Khan, K. R. (2020). Psychometric development and validation of attitude rating scale towards women empowerment: across male and female university population in Pakistan. International Journal of Human Rights in Healthcare.

- I recommend further improving the references by citing some of these recent studies on the topic:

Naeem, B., Aqeel, M., & de Almeida Santos, Z. (2021). Marital conflict, self-silencing, dissociation, and depression in married madrassa and non-madrassa women: a multilevel mediating model. Nature-Nurture Journal of Psychology, 1(2), 1-11.

Naeem, B., & Chaman, A. The Association of Adverse Self-Silencing and Marital Conflict with Symptoms of Depression and Dissociation in Married Madrassa and Non-Madrassa Women: A Cross-sectional Study.

Naeem, B. Nurturing the Soul: A Psychometric Analysis of the Spiritual Intelligence Inventory in Married Madrassa and Non-Madrassa Women.

Saif, J., Rohail, D. I., & Aqeel, M. (2021). Quality of Life, Coping Strategies, and Psychological Distress in Women with Primary and Secondary Infertility; A Mediating Model . Nature-Nurture Journal of Psychology, 1(1 SE-), 8–17.

Naeem, B., Aqeel, M., & de Almeida Santos, Z. (2021). Marital Conflict, Self-Silencing, Dissociation, and Depression in Married Madrassa and Non-Madrassa Women: A Multilevel Mediating Model. Nature-Nurture Journal of Psychology, 1(2), 1–11

Hafsa, S., Aqeel, M., & Shuja, K. H. (2021). The Moderating Role of Emotional Intelligence between Inter-Parental Conflicts and Loneliness in Male and Female Adolescents. Nature-Nurture Journal of Psychology, 1(1 SE-), 38–48

Reviewer #2: The paper cannot be considered for publication unless it is rewritten in proper English. Methodology and results sections sound good but there are so many linguistic errors that makes full understanding the paper not possible.

6. PLOS authors have the option to publish the peer review history of their article (what does this mean?). If published, this will include your full peer review and any attached files.

**Do you want your identity to be public for this peer review?** For information about this choice, including consent withdrawal, please see our Privacy Policy.

Reviewer #1: **Yes: **Muhammad Aqeel

Reviewer #2: No

---

## [Editor Report · Decision Letter 1]

11 Jun 2024

Assessing Africa’s child survival gains and prospects for attaining SDG target on child mortality

PGPH-D-23-02289R1

Dear Dr. Adedini,

We are pleased to inform you that your manuscript 'Assessing Africa’s child survival gains and prospects for attaining SDG target on child mortality' has been provisionally accepted for publication in PLOS Global Public Health.

Best regards,

Shaonong Dang, PhD

Academic Editor